# Factors Correlating to the Development of Hepatitis C Virus Infection in Hemodialysis Patients—Findings Mainly from Asiatic Populations: A Systematic Review and Meta-Analysis

**DOI:** 10.3390/ijerph16081453

**Published:** 2019-04-24

**Authors:** Gaofeng Cai, Hongjie Zheng, Lan Luo, Zhengting Wang, Zhenggang Jiang, Shuangfei Xu, Huakun Lv, Yongdi Chen, Biao Zhou, Chonggao Hu

**Affiliations:** 1Zhejiang Provincial Center for Disease Control and Prevention, 3399 Binsheng Road, Hangzhou 310051, China; gfcai@cdc.zj.cn (G.C.); ztwang@cdc.zj.cn (Z.W.); zhgjiang@cdc.zj.cn (Z.J.); sfxu@cdc.zj.cn (S.X.); hklv@cdc.zj.cn (H.L.); bzhou@cdc.zj.cn (B.Z.); chghu@cdc.zj.cn (C.H.); 2College of Medical and Technology, Zhejiang Chinese Medical University, Hangzhou 310051, China; 15868822853@163.com (H.Z.); luolanmmmm@163.com (L.L.); 3Zhejiang provincial key laboratory of infectious disease vaccine and prevention and control, 3399 Binsheng Road, Hangzhou 310051, China

**Keywords:** hemodialysis, hepatitis C virus, risk factor, meta-analysis

## Abstract

Hemodialysis is an effective replacement therapy for chronic renal failure patients. In recent decades, the number of hemodialysis patients has grown rapidly and some measures for preventing blood-borne diseases have been implemented, but hepatitis C virus (HCV) infection remains a significant problem. The meta-analysis published in 2009 on HCV infection-related factors was based on localized study objects, and some additional studies have been published since 2009; however, the contribution of these factors remains under dispute. Our study pooled the odds ratios (ORs) or mean standard deviations (MDs) with 95% confidence intervals (CIs) and analyzed sensitivity using Review Manager 5.1 software (5.1 version Copenhagen: The Nordic Cochrane Centre; 2011) by searching data in the PubMed, Elsevier, Springer, Wiley, and EBSCO databases. Spearman correlation analysis was performed using the SPSS package. In our meta-analysis, 1715 HCV-infected hemodialysis patients and 7093 non-HCV-infected hemodialysis patients from 44 studies were analyzed. The pooled ORs with 95% CIs were: histories of blood transfusion, 4.30 (3.11, 5.96); weekly hemodialysis times > 2, 6.00 (3.25, 11.06); kidney transplantation, 5.80 (3.95, 8.52); hemodialysis units > 2, 6.90 (2.42, 19.68); shared hemodialysis devices, 5.00 (2.35, 10.65); and drug addiction, 4.73 (1.54, 14.47). The pooled MDs with 95% CIs were duration of hemodialysis (months) 27.48 (21.67, 33.30). There was a positive correlation between duration of hemodialysis and the HCV infection rate (*p* < 0.01). Hemodialysis patients, especially from Asia, with shared hemodialysis devices, hemodialysis units > 2, blood transfusion, kidney transplantation, and drug addiction were at increased risk of HCV infection. The HCV infection rate increased with the duration of hemodialysis. High-risk hemodialysis patients should be monitored and receive timely screening.

## 1. Introduction

Hemodialysis is an effective replacement therapy for chronic renal failure patients that can increase survival times [1]. In recent decades, the number of patients administered hemodialysis has grown rapidly. In 2012, a study showed that an estimated 2.1 million patients needed hemodialysis therapy and that this number would increase by 7% annually on a global scale [2].

Hemodialysis patients are at high risk of blood-borne diseases, especially infectious hepatitis C virus (HCV) and hepatitis B virus (HBV) [3,4,5]. The infection rate of HBV and HCV in hemodialysis patients is dramatically different in different regions of the world; for example, the infection rate in the UK was reported to be 1%, compared with more than 90% in Eastern Europe [6]. In China, a multi-center study by Zhuang (2000) showed that the HCV infection rate was 16.3%–32.1% among hemodialysis patients [7], and another multi-center meta-analysis by Sun et al. (2009) showed a median rate of HCV infection of 41.10% for Chinese hemodialysis patients [8].

However, the rate and number of HBV infections have declined greatly since the World Health Organization recommended that HBV vaccination be included in national immunization programs in 1992, and subsequently mass vaccination for hepatitis B has been implemented on a population-wide scale [9,10,11,12].

By contrast, an effective vaccine is lacking for HCV, and hemodialysis patients who require regular invasive treatment and are at high risk of HCV infection [13]. A study by Abdulkarim et al. showed that the main cause of liver disease among hemodialysis patients was HCV infection [14]. In a large-scale clinical study by Tanaka et al., HCV-infected hemodialysis patients had a higher mortality rate compared with hemodialysis patients without HCV infection [15]. Furthermore, a study by Alter et al. showed that HCV-infected hemodialysis patients had a lower quality of life [16] and another study by Za et al. reported that the HCV-infected population had a higher incidence of liver fibrosis and liver cancer [17].

Over the past 10 years, most countries have published standard operating procedures for blood purification including measures such as using a dialyzer only once [18], dialyzing HCV-infected patients in a designated area [19,20], as well as introducing more stringent disinfection procedures and improving screening methods for donors [21].

However, although the rate of HCV infection among patients has declined, HCV infection among hemodialysis patients remains a significant problem [4,20,22].

Previous studies have shown that many factors influence the development of HCV infection, such as blood transfusion, duration on hemodialysis, weekly hemodialysis times, and a history of kidney transplantation [23,24,25,26,27,28,29,30,31,32,33,34,35,36,37,38,39,40]. In 2009, Sun et al. performed meta-analyses on Chinese hemodialysis patients, along with a limited subgroup analysis [8]. Some additional studies on risk factors of HCV infection in hemodialysis patients have also been published since 2009 [41,42,43,44,45,46,47,48,49,50,51,52,53,54,55,56,57,58,59,60,61,62,63]. However, the contribution of each of these factors is under dispute.

In this study, we used hemodialysis patients as research objects and extended the analysis of risk factors for HCV infection by performing a meta-analysis.

## 2. Materials and Methods

### 2.1. Literature Search Strategy

Searches were performed for each of the specified databases on the Bo Ku data service platform. We used the search terms “Hepatitis C”, “HCV”, and “Hemodialysis” in the search field “Title/Abstract”, and the electronic databases searched included the following six international databases: PubMed, Elsevier, Springer, Wiley, OVID, and EBSCO. We also used the search terms “Hepatitis C”, “HCV”, and “Hemodialysis” in the search field “Abstract” and searched two Chinese electronic databases: Chinese Medical Journal Database and Chinese National Knowledge Infrastructure. The searches were completed in the first week of September, 2018. We also searched the references listed at the end of the included articles.

### 2.2. Inclusion and Exclusion Criteria

In this study, the eligibility criteria for the inclusion of literature in the meta-analysis were as follows: (1) the literature is the original research; (2) the literature was an observational study with specific temporal and geographic characteristics; (3) the literature was published with the full text available; (4) all cases and controls were hemodialysis patents and the source of samples was clearly stated; (5) hepatitis C was diagnosed according to the national diagnostic criteria that existed at that time [64] and possible risk factors were reported; and (6) the literature was published in Chinese or English.

Literature was excluded from the meta-analysis when (1) the data reported could not be used to calculate the odds ratio (OR) or mean standard deviation (MD) with 95% confidence interval (CI) for the main variable; (2) the literature duplicated the same research; (3) the literature used the same research objects; and (4) the literature was deemed to be of poor quality literature (based on Ebrahim et al.’s declaration, the number of items satisfied in the corresponding research type declaration was less than half of total items) [65,66].

### 2.3. Data Extraction

We used a pre-made form for data extraction, and then two trained reviewers assessed the literature one by one and completed the form. The following data were extracted from the qualified studies: first author, year of the study, location, the number of hemodialysis patients in the HCV-infected group and the non-HCV-infected group, sample size, male to female ratio, and age distribution for HCV infection development among hemodialysis patents.

Discrepancies between the assessment results obtained by the two reviewers were resolved by discussion and checking the original documents.

### 2.4. Sensitivity Analysis

The studies with the widest 95% CI for the OR or MD were omitted from the subgroup analysis for this factor, and the remaining studies were pooled and pooled MD_CI_ or pooled OR_CI_ values with 95% CIs were obtained for this study factor, and then this pooled MD_CI_ or pooled OR_CI_ was compared with the total pooled OR or pooled MD before omitting this study factor. The studies with the maximum weight were omitted from the subgroup analysis, and then pooled, and the pooled MD_weight_ or OR_weight_ values with 95% CI for this study factor were obtained and then this pooled MD_weight_ or pooled OR_weight_ was compared with the total pooled MD or pooled OR before omitting this study factor.

In this meta-analysis, subgroup analyses were used to determine the associations between different study factors and HCV infection, a sensitivity analysis was used to examine the reliability of the associations, and a funnel plot was used to examine publication bias. According to the manufacturer’s instructions, the normal ranges of values for serum alanine aminotransferase (ALT) are about 5–40 units per liter of serum. Based on the standard procedure reported in the instructions, HCV infection was confirmed when a serum sample tested positive for HCV antibodies.

### 2.5. Statistical Analysis

The OR or MD with 95% CI were taken as the main indicators in this meta-analysis. Review Manager 5.1 software (5.1 version Copenhagen: The Nordic Cochrane Centre; 2011) was used to analyze the fixed-effect model without heterogeneity or the random-effect model with heterogeneity, after the heterogeneity test. The heterogeneity among different studies for study factors was evaluated by Cochran’s chi-square test with a significance level α = 0.1 and I^2^ statistics. The OR or MD was not pooled when its number for the study factor was less than 4. In the meta-analysis, I^2^ statistics, ranging from 0% to 100%, were used to assess the levels of heterogeneity; values of 0%, 25%, 50%, 75%, and 100% were taken as no, low, medium, high, and significant heterogeneity, respectively [67]. In this meta-analysis, I^2^ ≤ 50% was accepted. Correlation analysis was performed using SPSS version 16 software (SPSS Inc., Chicago, IL, USA). Spearman correlation was used for ranked data, with α = 0.05 considered to indicate statistical significance.

## 3. Results

### 3.1. Literature Search

In this meta-analysis, a total of 44 research articles were included, and a flow chart of the literature selection process is shown in Figure 1.

Of the literature selected, 44 studies included 1715 HCV-infected hemodialysis patients and 7093 non-HCV-infected hemodialysis patients. The study characteristics, region, study type, number of HCV-infected hemodialysis patients and non-HCV-infected hemodialysis patients, study factors, sample size, male/female ratio, and mean participant age (years) are shown in Table 1.

The 11 study factors used to pool OR or MD with 95% CI were as follows: gender (15 studies, 867 cases, 3874 controls), age (16 studies, 953 cases, 4770 controls); a history of blood transfusion (28 studies, 820 cases, 3663 controls); weekly hemodialysis times > 2 (5 studies, 103 cases, 344 controls); a history of kidney transplantation (8 studies, 234 cases, 1507 controls); hemodialysis units > 2 (6 studies, 127 cases, 389 controls); shared hemodialysis devices (6 studies, 275 cases, 1358 controls); serum alanine aminotransferase (ALT) levels (abnormal) (6 studies, 280 cases, 1344 controls); drug addiction (4 studies, 86 cases, 639 controls); a history of surgery (7 studies, 161 cases, 1123 controls); and duration of hemodialysis (months) (28 studies, 940 cases, 4044 controls).

### 3.2. Results of Pooled ORs or MDs

In this meta-analysis, the pooled ORs and their 95% CIs for study factors were as follows: histories of blood transfusion, 4.30 (3.11, 5.96); weekly hemodialysis times > 2, 6.00 (3.25, 11.06); a history of kidney transplantation, 5.80 (3.95, 8.52); hemodialysis units > 2, 6.90 (2.42, 19.68); shared hemodialysis devices, 5.00 (2.35, 10.65); serum ALT levels (abnormal), 5.62 (2.35, 13.40); drug addiction, 4.73 (1.54, 14.47); a history of surgery, 1.98 (1.37, 2.85); and gender (male), 0.98 (0.83, 1.15).

In this meta-analysis, the pooled MDs and their 95% CIs for study factors were as follows: duration of hemodialysis (months), 27.48 (21.67, 33.30); and age (years), −0.3 (−2.29, 1.69).

The pooled ORs with their 95% CIs for study factors including histories of blood transfusion, weekly hemodialysis times > 2, kidney transplantation, serum ALT levels (abnormal), drug addiction, a history of surgery, hemodialysis units > 2, and shared hemodialysis devices are detailed in Figure 2, Figure 3, Figure 4 and Figure 5.

### 3.3. Results of Rank Correlation Analysis

In terms of the duration of hemodialysis, groups of patients that underwent hemodialysis for 1–2 years, 2–3 years, 3–5 years, 5–10 years, and >10 years had HCV infection rates of 8.57%, 17.99%, 23.68%, 56.76%, and 60.47%, respectively, and pooled ORs with 95% CIs of 0.09 (0.06, 0.13), 0.57 (0.34, 0.97), 1.50 (0.86, 2.61), 6.07 (4.45, 8.28), and 5.76 (2.89, 11.46), respectively. There was a positive correlation between the duration of hemodialysis and the HCV infection rate (r_spearman_ = 0.990, *p* < 0.01) and between the duration of hemodialysis and the pooled OR (r_spearman_ = 0.900, *p* = 0.037).

### 3.4. Results of Heterogeneity Evaluation

A heterogeneity test for pooled ORs with 95% CIs showed that variations among ORs for study factors including histories of blood transfusion, hemodialysis units > 2, shared hemodialysis devices, and serum ALT levels were statistically significant (*p* < 0.10). The effects of these factors were then pooled using the random-effect model, whereas weekly hemodialysis times, kidney transplantation, drug addiction, a history of surgery, and gender were pooled using the fixed-effect model (*p* > 0.10). A heterogeneity test for pooled MDs with 95% CIs showed that the variation among studies for the duration of hemodialysis (months) was statistically significant (*p* < 0.10). The effects were then pooled using the random-effect model. These results are detailed in Figure 2, Figure 3, Figure 4 and Figure 5.

### 3.5. Publication Bias

In the meta-analysis, a funnel plot of the articles including the duration of hemodialysis was symmetrical, with the axis of symmetry (MD = 0) being to the right of center, as detailed in Figure 6.

### 3.6. Sensitivity Analysis

In view of the reliability of the pooled ORs or MDs using the random-effect model for terms including histories of blood transfusion, shared hemodialysis devices, hemodialysis units > 2, serum ALT levels, and duration of hemodialysis (months), we omitted studies with the widest 95% CIs for the ORs and MD values, respectively, and pooled and acquired OR_CI_ and MD_CI_ values with the 95% CIs, and these pooled values were close to the respective pooled OR and MD values with 95% CIs, as detailed in Table 2.

In view of the reliability of pooled ORs using the random-effect model for terms including histories of blood transfusion, shared hemodialysis devices, hemodialysis units > 2, serum ALT levels, and duration of hemodialysis (months), we omitted studies with the highest weights, and pooled and acquired OR_weight_ or MD_weight_ values with 95% CIs, and these pooled values were close to the respective pooled OR or MD values with 95% CIs, as detailed in Table 2.

## 4. Discussion

This study showed that, for hemodialysis patients, the rate of HCV infection increased with the duration of hemodialysis treatment. This meta-analysis also found that hemodialysis patients with a duration of hemodialysis treatment >5 years and/or histories of blood transfusion and/or shared hemodialysis devices and/or hemodialysis units >2 and/or weekly hemodialysis times >2 and/or kidney transplantation and/or histories of surgery and/or drug addiction were at increased risk of developing HCV infection, whereas the age and gender of hemodialysis patients did not affect the risk of developing HCV infection.

Our study analyzed the rate of HCV infection among groups of patients with a duration of hemodialysis treatment of 1–2 years, 2–3 years, 3–5 years, 5–10 years, and >10 years, and the study found that the longer the duration of hemodialysis treatment, the higher the rate of HCV infection; this result was consistent with the results reported in the meta-analysis by Sun et al. in 2009 [8].

More specifically, our study showed that patients with a duration of hemodialysis treatment >5 years were at increased risk of developing HCV infection, whereas patients with a duration of hemodialysis treatment <5 years did not have an increased risk of developing HCV infection. This result was not consistent with the findings of the 2009 meta-analysis, which reported that patients with a duration of hemodialysis treatment >1 year were at increased risk of developing HCV infection [8]. These differences may reflect the implementation of effective management measures imposed by relevant healthcare organizations in recent years.

In our meta-analysis, the result of a quantitative analysis (Figure 4) also showed that the duration of hemodialysis for HCV-infected patients was 27.48 months longer than the duration of hemodialysis for non-HCV-infected patients. This finding was longer than the 15.41 months reported in the 2009 meta-analysis by Sun et al. [8], and this difference may also reflect the implementation of the above-mentioned effective management measures in recent years. Moreover, this result of the quantitative analysis was also consistent with those of qualitative analysis in this study (i.e., patients with a duration of hemodialysis treatment >5 years did show an increased risk of developing HCV infection, whereas patients with a duration of hemodialysis treatment < 5 years were not at higher risk).

In general, exposure to HCV-contaminated medical equipment or goods can increase the risk of HCV infection, and during the process of hemodialysis, patients have many possible opportunities for exposure to HCV-contaminated equipment or goods [68,69].

The findings of this meta-analysis showed that hemodialysis patients with a history of shared hemodialysis devices and/or hemodialysis units >2 and/or weekly hemodialysis times >2 and/or a duration of hemodialysis >5 years were at increased risk of HCV infection, and this may be related to the fact that these hemodialysis patients had more opportunities to be exposed to HCV- contaminated medical equipment, HCV-contaminated goods, or the HCV-contaminated hands of medical personnel, potentially leading to nosocomial infection.

A study by Alfurayh et al. confirmed the existence of nosocomial transmission in hemodialysis centers by sequence analysis [70]. Moreover, the findings of this meta-analysis showed that hemodialysis patients with a history of drug addiction were at increased risk of HCV infection and this may be related to the fact that these hemodialysis patients had shared HCV-contaminated needles and syringes, leading to cross infection. From what has been discussed above, we suggest that disposable goods, such as disposable dialysis dialyzers, disposable dialysis pipes, and so on, should be used to cut off cross infection during hemodialysis.

The findings of this meta-analysis also showed that kidney transplantation hemodialysis patients were at increased risk of HCV infection, and this may be related to the fact that these hemodialysis patients had taken immunosuppressants, which may have resulted in low lymphocyte activation following HCV infection [71]. Moreover, the findings of this meta-analysis also showed that hemodialysis patients with abnormal serum ALT levels were at increased risk of HCV infection, and this may be related to the chronological order of the development of abnormal elevated serum ALT levels, which could not be identified in the observational studies included, or the fact that these hemodialysis patients had disrupted normal liver structure and function, which may have resulted in low lymphocyte activation following HCV infection.

In general, ELISA was the routine method for screening blood donors for HCV infection, but molecular-based tests such as PCR are more sensitive diagnostic assays, and thus it is possible that some blood donors screened by traditional ELISA methods may have been HCV infectors [72,73,74]. This may explain our finding that hemodialysis patients with histories of blood transfusion were at higher risk of developing HCV infection. Thus, we suggest that blood donors and hemodialysis patient populations should be tested regularly with more sensitive PCR diagnostic assays.

Our meta-analysis also found that the age and gender of hemodialysis patients did not affect the risk of developing HCV infection, and this result was consistent with the findings of the 2009 meta-analysis [8].

The sensitivity analysis performed as part of our meta-analysis found that, after omitting studies with the widest 95% CIs for OR or MD values and studies with the maximum weight in subgroup analyses for the duration of hemodialysis treatment (months), histories of blood transfusion, shared hemodialysis devices, hemodialysis units >2, and abnormal serum ALT levels, the two overall effects were not reversed and the pooled OR or MD values were similar to those observed before omitting the studies. This revealed that the pooled ORs or MDs for these study factors were reliable and stable.

The limitations of this study were that only articles published in English or Chinese were included in the meta-analysis. In addition, even though the ORs or MDs of the six factors were pooled using a random-effect method, study heterogeneity may have influenced the findings to some extent. Furthermore, some study factors, for example, the degree of deterioration and the socioeconomic status of patients, were not available to be pooled. Lastly, studies included were carried out in the following countries: Iran, Australia, Egypt, and China, and this aspect limits the generalizability of conclusions.

## 5. Conclusions

It can be concluded that, for hemodialysis patients, the rate of HCV infection increases with the duration of hemodialysis treatment, and that hemodialysis patients, especially from Asia, with histories of blood transfusion and/or weekly hemodialysis times >2 and/or shared hemodialysis devices and/or hemodialysis units >2 and/or kidney transplantation and/or drug addiction were at increased risk of developing HCV infection. High-risk hemodialysis patients should be closely monitored and receive timely screening and therapeutic intervention to reduce the risk of HCV nosocomial infection.

## Figures and Tables

**Figure 1 ijerph-16-01453-f001:**
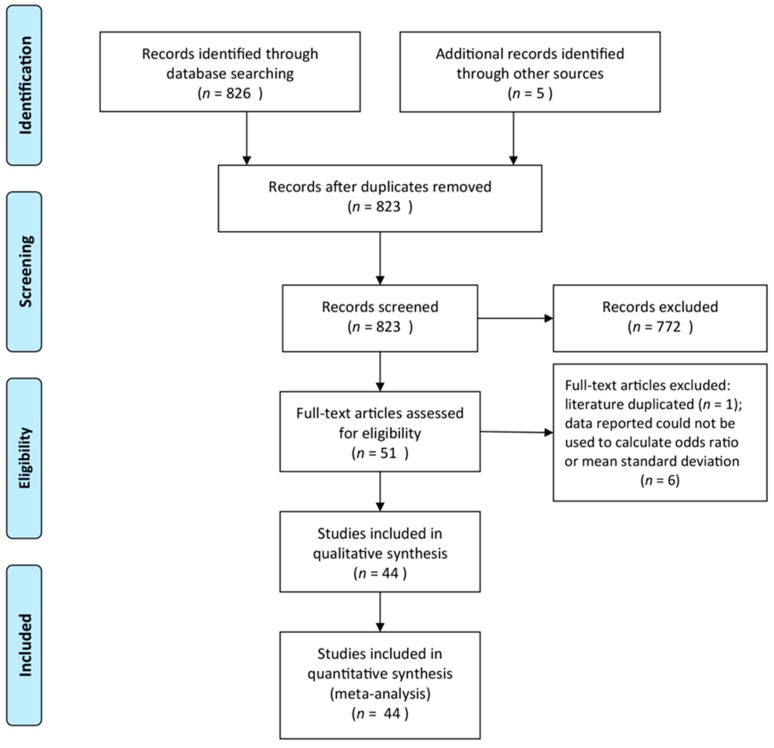
A flow chart of the literature selection process [68].

**Figure 2 ijerph-16-01453-f002:**
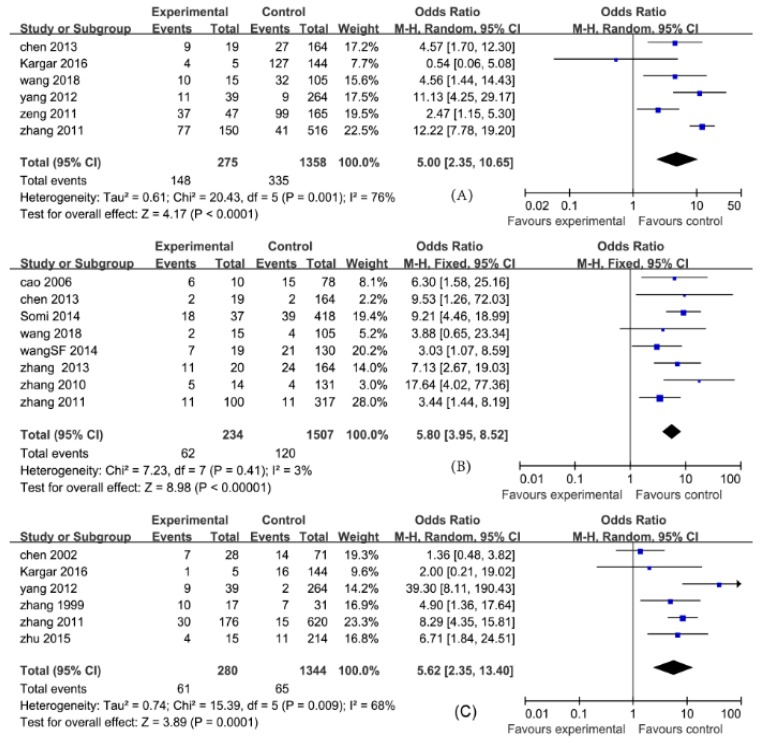
Effects of pooled odds ratios (ORs) for factors correlating to the development of hepatitis C virus infection in hemodialysis patients ((**A**) shared hemodialysis devices; (**B**) kidney transplantation; (**C)**: serum alanine aminotransferase levels (abnormal)).

**Figure 3 ijerph-16-01453-f003:**
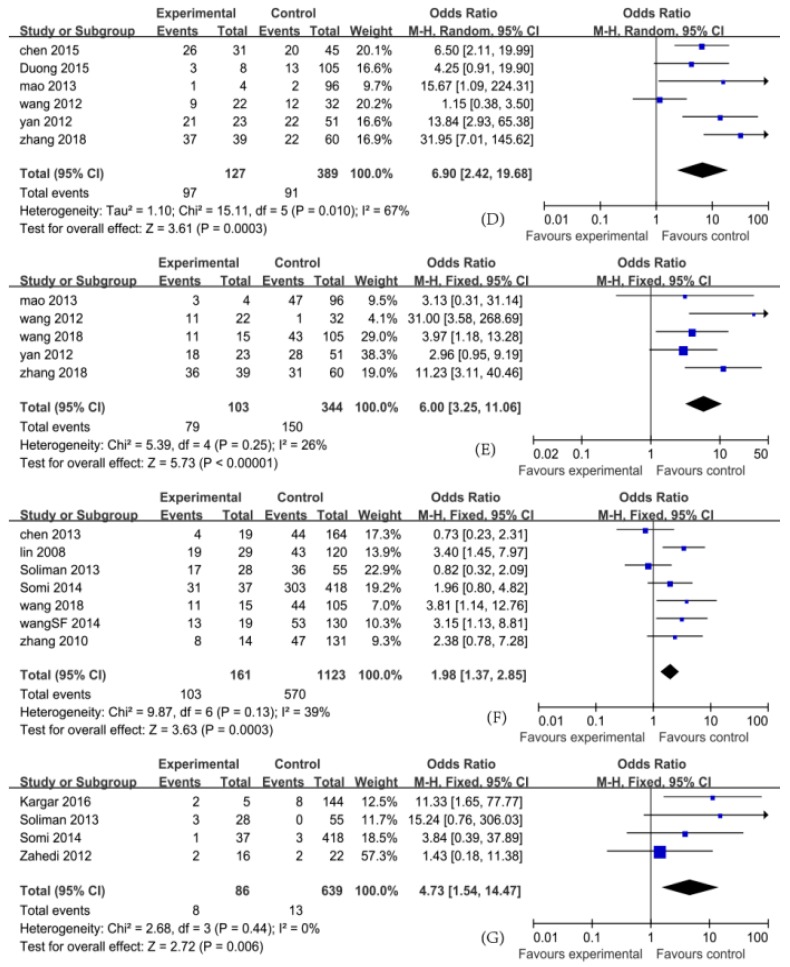
Effects of pooled ORs for factors correlating to the development of hepatitis C virus infection in hemodialysis patients ((**D**) hemodialysis units > 2; (**E**) weekly hemodialysis times > 2; (**F**) a history of surgery; (**G**) drug addiction).

**Figure 4 ijerph-16-01453-f004:**
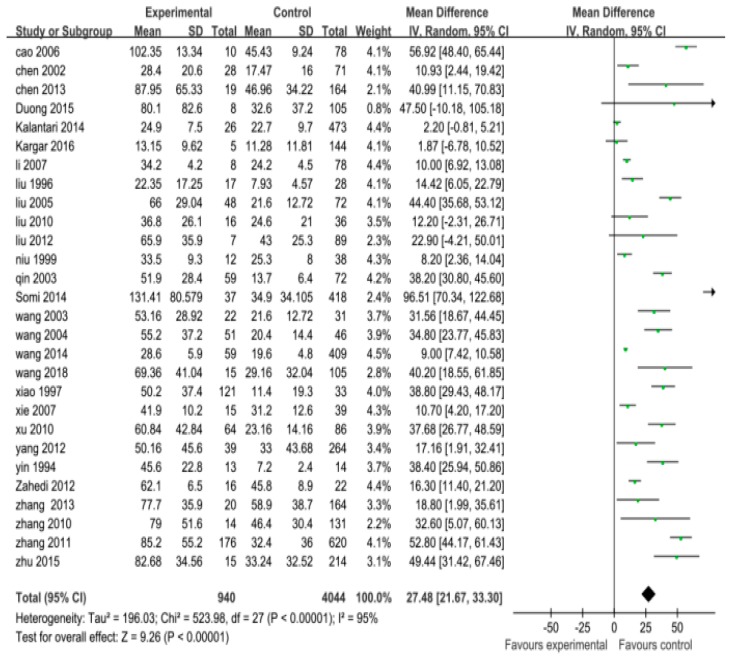
Effects of pooled mean standard deviations (MDs) for duration of hemodialysis (months) correlating to the development of hepatitis C virus infection in hemodialysis patients.

**Figure 5 ijerph-16-01453-f005:**
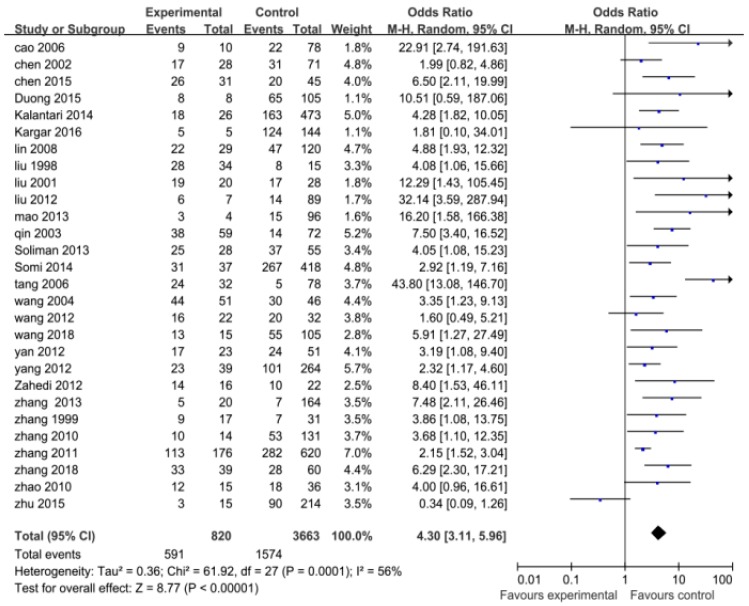
Effects of pooled OR for histories of blood transfusion correlating to the development of hepatitis C virus infection in hemodialysis patients.

**Figure 6 ijerph-16-01453-f006:**
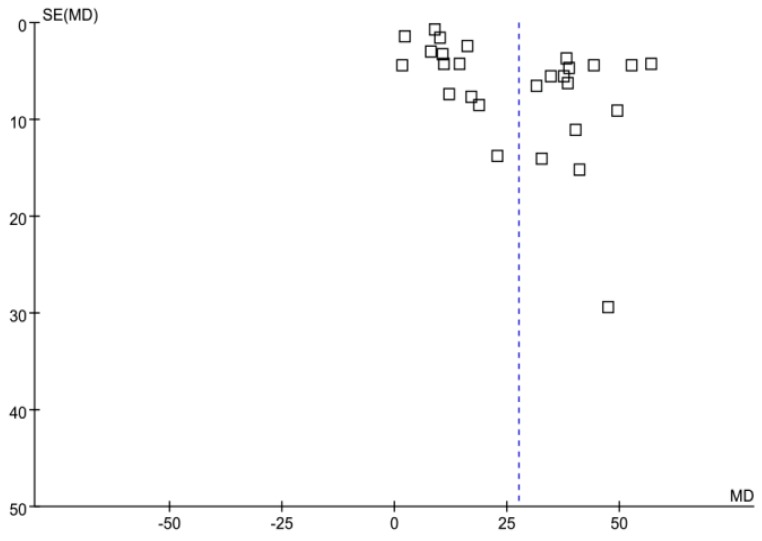
A funnel plot of the articles including the duration of hemodialysis.

**Table 1 ijerph-16-01453-t001:** Characteristics of the studies.

Reference Number	Author and Year of Publication	Regions	Study Type	Participants Category (Case/Control)	Sample Size (Case/Control)	Male/Female	Age (Years) *
4	Zahedi 2012	Iran, Kerman	observational study	HCV-infected hemodialysis patients/non-HCV-infected hemodialysis patients	16/22	24/14	51 ± 9
20	Chen 2013	Guangdong, Shenzhen	observational study	HCV-infected hemodialysis patients/non-HCV-infected hemodialysis patients	19/164	117/66	47.14 ± 15.196
22	Kargar 2016	Iran, Hormozgan	observational study	HCV-infected hemodialysis patients/non-HCV-infected hemodialysis patients	5/144	92/57	56.23 ± 12.35
23	Zhang 1999	China, Chongqing	observational study	HCV-infected hemodialysis patients/non-HCV-infected hemodialysis patients	17/31	30/18	50.5 ± 13.5
59	Duong 2015	Australia, Sydney	observational study	HCV-infected hemodialysis patients/non-HCV-infected hemodialysis patients	8/105	59/54	53 ± 16
60	Somi 2014	Iran, Tabriz	observational study	HCV-infected hemodialysis patients/non-HCV-infected hemodialysis patients	37/418	275/180	55.98 ± 15.6
61	Kalantari 2014	Iran, Isfahan	observational study	HCV-infected hemodialysis patients/non-HCV-infected hemodialysis patients	26/473	303/196	52.3 ± 12.8
62	Chang 2014	Taiwan, Kaohsiung	observational study	HCV-infected hemodialysis patients/non-HCV-infected hemodialysis patients	290/1391	824/857	62.2 ± 12.9
63	Soliman 2013	Egypt, Cairo	observational study	HCV-infected hemodialysis patients/non-HCV-infected hemodialysis patients	28/55	47/36	52.29 ± 12.10; 49.47 ± 15.5
24	Cao 2006	China, Beijing	observational study	HCV-infected hemodialysis patients/non-HCV-infected hemodialysis patients	10/78	42/88	52.33 ± 12.55; 57.45 ± 12.16
25	Chen 2002	China, Beijing	observational study	HCV-infected hemodialysis patients/non-HCV-infected hemodialysis patients	28/71	57/42	58.1 ± 13.2; 57.7 ± 12.9 1)
26	Liu 1996	China, Shenyang	observational study	HCV-infected hemodialysis patients/non-HCV-infected hemodialysis patients	17/28	33/12	23–62
27	Liu 1998	China, Beijing	observational study	HCV-infected hemodialysis patients/non-HCV-infected hemodialysis patients	34/15	21/28	53.1 ± 11.2
28	Liu 2005	Xinjiang, Shihe	observational study	HCV-infected hemodialysis patients/non-HCV-infected hemodialysis patients	48/72	80/40	19–81
29	Wang 2003	Anhui, Benyang	observational study	HCV-infected hemodialysis patients/non-HCV-infected hemodialysis patients	22/31	40/11	46.68 ± 10.18; 47.50 ± 13.44
30	Wang 2004	China, Shanahai	observational study	HCV-infected hemodialysis patients/non-HCV-infected hemodialysis patients	51/97	57/91	51 ± 14
31	Xiao 1997	Jiangsu, Nanji	observational study	HCV-infected hemodialysis patients/non-HCV-infected hemodialysis patients	121/33	110/44	46.0 ± 12.8
32	Zhao 2008	China, Dalin	observational study	HCV-infected hemodialysis patients/non-HCV-infected hemodialysis patients	124/583	_	60.25 ± 13.95; 63.51 ± 17.52
33	Li 2008	Fujian, Fuzhou	observational study	HCV-infected hemodialysis patients/non-HCV-infected hemodialysis patients	29/120	101/48	13–82
34	Qin 2003	China, Dalin	observational study	HCV-infected hemodialysis patients/non-HCV-infected hemodialysis patients	59/135	77/117	49.7 ± 16.4
35	Tang 2006	China, Shanahai	observational study	HCV-infected hemodialysis patients/non-HCV-infected hemodialysis patients	32/78	59/51	49.2 ± 14.3
36	Liu 2001	Shandong, Liaocheng	observational study	HCV-infected hemodialysis patients/non-HCV-infected hemodialysis patients	20/28	36/12	17–76
37	Niu 1999	Guangdong, Shenzhen	observational study	HCV-infected hemodialysis patients/non-HCV-infected hemodialysis patients	12/38	27/23	48.5 + 12.8; 49.4 + 16.8
38	Yin 1994	Jiangsu, Nanji	observational study	HCV-infected hemodialysis patients/non-HCV-infected hemodialysis patients	14/13	17/10	21–69
39	Xie 2007	Haikou, Hainan	observational study	HCV-infected hemodialysis patients/non-HCV-infected hemodialysis patients	15/39	36/18	48.4
40	Li 2007	Guangdong, Zhanjiang	observational study	HCV-infected hemodialysis patients/non-HCV-infected hemodialysis patients	8/78	46/40	49.3 (25–71)
41	Liu 2010	Jilin, Songyuan	observational study	HCV-infected hemodialysis patients/non-HCV-infected hemodialysis patients	16/36	32/20	48.7 ± 2.1
42	Liu 2012	Liaoning, Shenyang	observational study	HCV-infected hemodialysis patients/non-HCV-infected hemodialysis patients	7/89	42/54	52.8 ± 15.4
43	Mao 2013	Zhejiang, Taizhou	observational study	HCV-infected hemodialysis patients/non-HCV-infected hemodialysis patients	4/96	57/43	50.8 (25–72)
44	Wang 2012	Hunan, Zhuzhou	observational study	HCV-infected hemodialysis patients/non-HCV-infected hemodialysis patients	22/32	35/19	51.88 ± 13.10
45	Wang, S.F. 2014	Anhui, Hefei	observational study	HCV-infected hemodialysis patients/non-HCV-infected hemodialysis patients	19/130	1.33/1	52.6 ± 11.3
46	Wang 2018	China, Chongqing	observational study	HCV-infected hemodialysis patients/non-HCV-infected hemodialysis patients	15/105	69/51	57.86 ± 7.85
47	Wang, L.F. 2014	Zhejiang, Qunan	observational study	HCV-infected hemodialysis patients/non-HCV-infected hemodialysis patients	59/409	249/219	21–78
48	Xu 2010	Guangdong, Zhanjiang	observational study	HCV-infected hemodialysis patients/non-HCV-infected hemodialysis patients	64/86	80/70	17–84
49	Yan 2012	Hunan, Changsha	observational study	HCV-infected hemodialysis patients/non-HCV-infected hemodialysis patients	23/51	47/27	47.72 ± 18.93
50	Yang 2012	Jiangsu, Yangzhou	observational study	HCV-infected hemodialysis patients/non-HCV-infected hemodialysis patients	39/264	194/109	48.49 ± 11.45; 50.08 ± 12.95
51	Zeng 2011	Guangdong, Qingyuan	observational study	HCV-infected hemodialysis patients/non-HCV-infected hemodialysis patients	47/165	116/96	16–76
52	Zhang 2013	China, Beijing	observational study	HCV-infected hemodialysis patients/non-HCV-infected hemodialysis patients	20/164	97/87	55.0 ± 15.6
53	Zhang 2010	China, Beijing	observational study	HCV-infected hemodialysis patients/non-HCV-infected hemodialysis patients	14/131	_	_
54	Zhu 2015	Jiangsu, Changzhou	observational study	HCV-infected hemodialysis patients/non-HCV-infected hemodialysis patients	15/214	134/95	56.25 ± 13.38; 51.04 ± 13.20
55	Zhang 2018	Hebei, Shijiazhuang	observational study	HCV-infected hemodialysis patients/non-HCV-infected hemodialysis patients	39/60	74/25	54.58 ± 11.88
56	Chen 2015	Hubei, Enshi	observational study	HCV-infected hemodialysis patients/non-HCV-infected hemodialysis patients	31/45	43/33	53.21 ± 10.61
57	Zhang 2011	Jiangsu, Nanji	observational study	HCV-infected hemodialysis patients/non-HCV-infected hemodialysis patients	176/620	492/304	53 ± 12; 52 ± 14
58	Zhao 2010	Shandong, Weifang	observational study	HCV-infected hemodialysis patients/non-HCV-infected hemodialysis patients	15/36	27/24	l9–78

Note: HCV: hepatitis C virus; *: mean ± standard deviation; mean (minimum–maximum); minimum–maximum; mean.

**Table 2 ijerph-16-01453-t002:** The subgroup characteristics of study factors associated with HCV infection in hemodialysis patients after omitting the studies with maximum value of weight or widest interval of 95% CI for OR or MD values in subgroup analysis.

Subgroup Analyses by Study Factors	OR or MD(95% CI) before Reference Omitted	OR or MD (95% CI) after Reference Omitted	Reversal of OR or MD (95% CI) after Reference Omitted Compared with that before Reference Omitted	Reference Omitted
The studies with wide interval of 95% CI for OR values				
Histories of blood transfusion	4.30 (3.11–5.96)	5.13 (3.43–7.68)	No	42
Shared hemodialysis devices	5.00 (2.35–10.65)	4.10 (1.66–10.16)	No	50
Hemodialysis units > 2	6.90 (2.42–19.68)	6.35 (2.01–20.08)	No	43
Serum alanine aminotransferase levels	5.62 (2.35–13.40)	4.17 (1.89–9.23)	No	50
The studies with wide interval of 95% CI for MD values				
Duration of hemodialysis (months)	28.96 (22.11–35.80)	23.38 (18.17–28.59)	No	59
The studies with maximum value of weight (OR)				
Histories of blood transfusion	4.30 (3.11–5.96)	4.45 (3.26–6.09)	No	57
Shared hemodialysis devices	5.00 (2.35–10.65)	3.97 (1.89–8.32)	No	57
Hemodialysis units > 2	6.90 (2.42–19.68)	10.16 (4.95–20.85)	No	44
Serum alanine aminotransferase levels	5.62 (2.35–13.40)	5.06 (1.60–15.98)	No	57
The studies with maximum value of weight (MD)				
Duration of hemodialysis (months)	28.96 (22.11–35.80)	26.36 (19.53–33.19)	No	61,47

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
