# Peer review of "Factors Correlating to the Development of Hepatitis C Virus Infection in Hemodialysis Patients—Findings Mainly from Asiatic Populations: A Systematic Review and Meta-Analysis"

_ijerph, 2019, doi:10.3390/ijerph16081453_

Round 1

Reviewer 1 Report

I have had  big difficulties in giving an unbiased judgement on this paper.

In fact  the informative value of this study is consistently limited by the  choice of the included references on which the authors built up their metanalysis. In fact, the quoted papers are almost totally  related to studies carried out in Asian  Countries and this aspects limits a lot the generalizability of the metanalysis. Furthermore , many statements are sometimes m very expected or not understandable( e.g. increased ALT levels are reported as a risk factor for developing  HCV infection ?!?). There are also many grammar and typing errors.

Author Response

Dear Reviewer:

We would like to take the opportunity to thank you all for your time and efforts. Your professional comments and suggestions for revision are very much appreciated. We have responded to the comments point by point.

Please see the comments and responses below and find the revisions in the manuscript.

Regards,

Yongdi Chen on behalf of all the authors

Point 1: I have had big difficulties in giving an unbiased judgement on this paper. In fact the informative value of this study is consistently limited by the choice of the included references on which the authors built up their metanalysis. In fact, the quoted papers are almost totally related to studies carried out in Asian Countries and this aspects limits a lot the generalizability of the meta-nalysis.

Response: Thank you for your vital comments. In this meta-analysis, given the study design and databases searched restriction(PubMed, Elsevier, Springer, Wiley, OVID, EBSCO, Chinese Medical Journal Database and Chinese National Knowledge Infrastructure), we included 44 eligibility studies carried out in Countries: Iran, Australia, Egypt and China, etc. (Table 1), and this aspects limits the generalizability of this meta-nalysis. We identified this bias, and this bias was stated in the limitations and the conclusions were targeted towards the studied populations. We add this argument and revised as follows:

Page 14, line 310-312. For the last one, studies included were carried out in Countries: Iran, Australia, Egypt and China, and this aspect limits the generalizability of conclusions.

Page 15, line 315-320. It can be concluded that, for hemodialysis patients, the rate of HCV infection increases with the duration of hemodialysis treatment, and that hemodialysis patients, especially from Asia and the Middle East, with a history of blood transfusion and/or weekly hemodialysis times >2 and/or shared hemodialysis devices and/or hemodialysis units >2 and/or kidney transplantation and/or drug addiction were at increased risk of developing HCV infection.

Point 2: Furthermore, many statements are sometimes m very expected or not understandable ( e.g. increased ALT levels are reported as a risk factor for developing  HCV infection ?)

Responses: Thank you for your professional suggestions for revision. We neglected to give an explanation. Now, we add an explanation in our discussion and revised as follows:

Page 14, line 283-289. In this meta-analysis, the findings that hemodialysis patients with abnormal serum ALT levels were at increased risk of HCV infection, this may be related to the fact chronological order of the development of abnormal elevated serum ALT levels and studies carried out couldn’t be identified in observational study included, or the fact that these hemodialysis patients had disrupted normal liver structure and function, that resulted in low lymphocyte activation following HCV infection.

Point 3: There are also many grammar and typing errors.

Responses: Thank you very much for your reminding. We have got a native speaker of English editing this manuscript. 

Reviewer 2 Report

I would first comment that these findings are fairly expected, but the overall study is very well done. There is a bias though that has to do with the majority of articles represented are from Asian/middle eastern centers, so their findings are most applicable to these populations this should be clearly stated. Why there are so few western sources included was not explicitly mentioned-but matters little, all that matters is identifying this bias so that the conclusions are targeted towards the studied populations.

Further, it should also be stated that patients with the exposures leading to a higher risk of HepC should be promptly screened not just "treated" as without timely screening, treatment would not be able to be targeted to infected patients.

Infection control measures, and measures to reduce Hcv transmission it seems are clearly needed in the aforementioned medical environments. The paper methodology is standard and the statistics are sensible and the conclusions are straight forward. I would only wish the authors would devise more concrete detection strategies for at risk populations, as well as more concrete comments about infection control.

Author Response

Dear Reviewer:

We would like to take the opportunity to thank you all for your time and efforts. Your professional comments and suggestions for revision are very much appreciated. We have responded to the comments point by point.

Please see the comments and responses below and find the revisions in the manuscript.

Regards,

Yongdi Chen on behalf of all the authors

Point 1: I would first comment that these findings are fairly expected, but the overall study is very well done. There is a bias though that has to do with the majority of articles represented are from Asian/middle eastern centers, so their findings are most applicable to these populations this should be clearly stated. Why there are so few western sources included was not explicitly mentioned-but matters little, all that matters is identifying this bias so that the conclusions are targeted towards the studied populations.

Response: Thank you for your vital comments. In this meta-analysis, given the study design and databases searched restriction(PubMed, Elsevier, Springer, Wiley, OVID, EBSCO, Chinese Medical Journal Database and Chinese National Knowledge Infrastructure), we included 44 eligibility studies carried out in Countries: Iran, Australia, Egypt and China, etc. (Table 1), and this aspects limits the generalizability of this meta-nalysis. We identified this bias, and this bias was stated in the limitations and the conclusions were targeted towards the studied populations. We added this argument and revised as follows:

Page 14, line 310-312. For the last one, studies included were carried out in Countries: Iran, Australia, Egypt and China, and this aspect limits the generalizability of conclusions.

Page 15, line 315-320. It can be concluded that, for hemodialysis patients, the rate of HCV infection increases with the duration of hemodialysis treatment, and that hemodialysis patients, especially from Asia and the Middle East, with a history of blood transfusion and/or weekly hemodialysis times >2 and/or shared hemodialysis devices and/or hemodialysis units >2 and/or kidney transplantation and/or drug addiction were at increased risk of developing HCV infection.

Point 2: Further, it should also be stated that patients with the exposures leading to a higher risk of Hep C should be promptly screened not just "treated" as without timely screening, treatment would not be able to be targeted to infected patients. Infection control measures, and measures to reduce Hcv transmission it seems are clearly needed in the aforementioned medical environments. The paper methodology is standard and the statistics are sensible and the conclusions are straight forward. I would only wish the authors would devise more concrete detection strategies for at risk populations, as well as more concrete comments about infection control.

Responses: Thank you for your professional suggestions for revision. We added screening and more concrete detection strategies for at risk populations, as well as more concrete comments about infection control in our discussion. we revised as follows:

Page 14, line 290-295. In general, ELISA was the routine method for screening blood donors for HCV infection, but molecular-based tests such as PCR were more sensitive diagnostic assays, and thus, it was possible that some donors screened by traditional ELISA methods may be HCV infectors [72-74]. This may explain our finding that hemodialysis patients with histories of blood transfusion were at higher risk of developing HCV infection. Thus, we devise that risk populations should be tested regularly with more sensitive PCR diagnostic assays.

Page14, line 271-278. A study by Alfurayh et al. confirmed the existence of nosocomial transmission in hemodialysis centers by sequence analysis [70]. Moreover, the findings of this meta-analysis showed that hemodialysis patients with a history of drug addiction were at increased risk of HCV infection and this may be related to the fact that these hemodialysis patients had shared HCV-contaminated needles and syringes, leading to cross-infection. From what has been discussed above, we devise that disposable goods, such as disposable dialysis dialyzers, disposable dialysis pipes, and so on, should be used to cut off cross infection during hemodialysis.

Reviewer 3 Report

Interesting paper, well written abstract and background, but the methods need some editing, and acronyms should be spelled out (i.e. MDCI and ORCI). Also, I didn't understand what 'the literature is the origin document' meant (under Exclusion Criteria). Line 97 describes the 'national diagnostic criteria', this should be defined in the paper and referenced. Figure 1 is missing a footnote and the figure is hard to read, perhaps a better quality image could be used. Figures 2-3 should be labeled with the appropriate letter A - G so that the reader knows which factor is represented. Table 2 is confusing; perhaps it could be made clearer what is being compared to what, and what the comparison means. Line 252 describes the increase in months of duration of hemodialysis comparing HCV to non-HCV infected patients, but I couldn't find this in the results.  Finally, it's clear from Table 1 that the vast majority of the studies were from China, which limits the ability to generalize to other countries. This should be stated in the limitations.

Author Response

Dear Reviewer:

We would like to take the opportunity to thank you all for your time and efforts. Your professional comments and suggestions for revision are very much appreciated. We have responded to the comments point by point.

Please see the comments and responses below and find the revisions in the manuscript.

Regards,

Yongdi Chen on behalf of all the authors

Point 1: Interesting paper, well written abstract and background, but the methods need some editing, and acronyms should be spelled out (i.e. MDCI and ORCI). Also, I didn't understand what 'the literature is the origin document' meant (under Exclusion Criteria).

Responses: Thank you very much for the vital comments, and we have made the corrections accordingly as follows:

1)      acronyms have be spelled out (i.e. MDCI and ORCI) in Page 3, line 102-103.

2)      We have corrected ‘the literature is the origin document’ into ‘the literature is the origin research’ in Page 3, line 95.

Point 2: Line 97 describes the 'national diagnostic criteria', this should be defined in the paper and referenced.

Responses: 'National diagnostic criteria'have been defined and referenced (No. 64), in Page 3, Line 99

Point 3:Figure 1 is missing a footnote and the figure is hard to read, perhaps a better quality image could be used. Figures 2-3 should be labeled with the appropriate letter A - G so that the reader knows which factor is represented.

Responses: We have added a footnote in Figure 1 and a better quality image have been used. Figures 2-3 have been labeled with the appropriate letter A - G.

Point 4: Table 2 is confusing; perhaps it could be made clearer what is being compared to what, and what the comparison means.

Responses:Table 2 have been corrected. Table 2 could be made clearer what is being compared to what, and what the comparison means.

Point 5: Line 252 describes the increase in months of duration of hemodialysis comparing HCV to non-HCV infected patients, but I couldn't find this in the results.

Responses: Line 252 describes the increase in 27.48 months of duration of hemodialysis comparing HCV to non-HCV infected patients, please see Total (95%CI) (Figure 4).

Point 6: Finally, it's clear from Table 1 that the vast majority of the studies were from China, which limits the ability to generalize to other countries. This should be stated in the limitations.

Responses: Thank you for your vital comments. In this meta-analysis, given the study design and databases searched restriction(PubMed, Elsevier, Springer, Wiley, OVID, EBSCO, Chinese Medical Journal Database and Chinese National Knowledge Infrastructure), we included 44 eligibility studies carried out in Countries: Iran, Australia, Egypt and China, etc. (Table 1), and this aspects limits the generalizability of this meta-nalysis. We identified this bias, and this bias was stated in the limitations and the conclusions were targeted towards the studied populations. We added this argument and revised as follows:

Page 14, line 310-312. For the last one, studies included were carried out in Countries: Iran, Australia, Egypt and China, and this aspect limits the generalizability of conclusions.

Page 15, line 315-320. It can be concluded that, for hemodialysis patients, the rate of HCV infection increases with the duration of hemodialysis treatment, and that hemodialysis patients, especially from Asia and the Middle East, with a history of blood transfusion and/or weekly hemodialysis times >2 and/or shared hemodialysis devices and/or hemodialysis units >2 and/or kidney transplantation and/or drug addiction were at increased risk of developing HCV infection.

Round 2

Reviewer 1 Report

though the authors made a certain effort for improving the manuscript, the main limitations described in the first revision are still present.

So, I think that at least the title should be refrased specifying  that the present metanalysis is mainly focused on the  asiatic populations

Author Response

Dear Reviewer:

We would like to take the opportunity to thank you all for your time and efforts. Your rigorous scientific attitude and suggestions for revision are very much appreciated. We have responded to the comments point by point.

Please see the comments and responses below and find the revisions in the manuscript (in red).

Regards,

Yongdi Chen on behalf of all the authors

Point 1: though the authors made a certain effort for improving the manuscript, the main limitations described in the first revision are still present.

So, I think that at least the title should be refrased specifying that the present metanalysis is mainly focused on the asiatic populations

Response 1: Thank you for your vital comments. We have revised the title as follows: Factors correlating to the development of hepatitis C virus infection in hemodialysis patients - findings mainly from Asiatic populations: a systematic review and meta-analysis

Please see the revised manuscript: line 3 – 4, line 35 and line 317.   
